# TRACE NORM REGULARIZATION AND FASTER INFERENCE FOR EMBEDDED SPEECH RECOGNITION RNNS

## ABSTRACT

We propose and evaluate new techniques for compressing and speeding up dense matrix multiplications as found in the fully connected and recurrent layers of neural networks for embedded large vocabulary continuous speech recognition (LVCSR). For compression, we introduce and study a trace norm regularization technique for training low rank factored versions of matrix multiplications. Compared to standard low rank training, we show that our method leads to good accuracy versus number of parameter trade-offs and can be used to speed up training of large models. For speedup, we enable faster inference on ARM processors through new open sourced kernels optimized for small batch sizes, resulting in 3x to 7x speed ups over the widely used gemmlowp library. Beyond LVCSR, we expect our techniques and kernels to be more generally applicable to embedded neural networks with large fully connected or recurrent layers.

## 1 INTRODUCTION

For embedded applications of machine learning, we seek models that are as accurate as possible given constraints on size and on latency at inference time. For many neural networks, the parameters and computation are concentrated in two basic building blocks:

1. **Convolutions**. These tend to dominate in, for example, image processing applications.

2. **Dense matrix multiplications** (GEMMs) as found, for example, inside fully connected layers or recurrent layers such as GRU and LSTM. These are common in speech and natural language processing applications.

These two building blocks are the natural targets for efforts to reduce parameters and speed up models for embedded applications. Much work on this topic already exists in the literature. For a brief overview, see Section 2.

In this paper, we focus only on dense matrix multiplications and not on convolutions. Our two main contributions are:

1. **Trace norm regularization:** We describe a trace norm regularization technique and an accompanying training methodology that enables the practical training of models with competitive accuracy versus number of parameter trade-offs. It automatically selects the rank and eliminates the need for any prior knowledge on suitable matrix rank.

2. **Efficient kernels for inference:** We explore the importance of optimizing for low batch sizes in on-device inference, and we introduce kernels[1] for ARM processors that vastly outperform publicly available kernels in the low batch size regime.

These two topics are discussed in Sections 3 and 4, respectively. Although we conducted our experiments and report results in the context of large vocabulary continuous speech recognition (LVCSR) on embedded devices, the ideas and techniques are broadly applicable to other deep learning networks. Work on compressing any neural network for which large GEMMs dominate the parameters or computation time could benefit from the insights presented in this paper.

---

[1] Available at `https://github.com/paddlepaddle/farm`.

## 2 RELATED WORK

Our work is most closely related to that of Prabhavalkar et al. (2016), where low rank factored acoustic speech models are similarly trained by initializing weights from a truncated singular value decomposition (SVD) of pretrained weight matrices. This technique was also applied to speech recognition on mobile devices (McGraw et al., 2016; Xue et al., 2013). We build on this method by adding a variational form of trace norm regularization that was first proposed for collaborative prediction (Srebro et al., 2005) and also applied to recommender systems (Koren et al., 2009). The use of this technique with gradient descent was recently justified theoretically (Ciliberto et al., 2017). Furthermore, Neyshabur et al. (2015) argue that trace norm regularization could provide a sensible inductive bias for neural networks. To the best of our knowledge, we are the first to combine the training technique of Prabhavalkar et al. (2016) with variational trace norm regularization.

Low rank factorization of neural network weights in general has been the subject of many other works (Denil et al., 2013; Sainath et al., 2013; Ba & Caruana, 2014; Kuchaiev & Ginsburg, 2017). Some other approaches for dense matrix compression include sparsity (LeCun et al., 1989; Narang et al., 2017), hash-based parameter sharing (Chen et al., 2015), and other parameter-sharing schemes such as circulant, Toeplitz, or more generally low-displacement-rank matrices (Sindhwani et al., 2015; Lu et al., 2016). Kuchaiev & Ginsburg (2017) explore splitting activations into independent groups. Doing so is akin to using block-diagonal matrices.

The techniques for compressing convolutional models are different and beyond the scope of this paper. We refer the interested reader to, e.g., Denton et al. (2014); Han et al. (2016); Iandola et al. (2016) and references therein.

## 3 TRAINING LOW RANK MODELS

Low rank factorization is a well studied and effective technique for compressing large matrices. In Prabhavalkar et al. (2016), low rank models are trained by first training a model with unfactored weight matrices (we refer to this as stage 1), and then initializing a model with factored weight matrices from the truncated SVD of the unfactored model (we refer to this as *warmstarting* a stage 2 model from a stage 1 model). The truncation is done by retaining only as many singular values as required to explain a specified percentage of the variance.

If the weight matrices from stage 1 had only a few nonzero singular values, then the truncated SVD used for warmstarting stage 2 would yield a much better or even error-free approximation of the stage 1 matrix. This suggests applying a sparsity-inducing $\ell^1$ penalty on the vector of singular values during stage 1 training. This is known as trace norm regularization in the literature. Unfortunately, there is no known way of directly computing the trace norm and its gradients that would be computationally feasible in the context of large deep learning models. Instead, we propose to combine the two-stage training method of Prabhavalkar et al. (2016) with an indirect variational trace norm regularization technique (Srebro et al., 2005; Ciliberto et al., 2017). We describe this technique in more detail in Section 3.1 and report experimental results in Section 3.2.

### 3.1 TRACE NORM REGULARIZATION

First we introduce some notation. Let us denote by $||\cdot||_\mathcal{T}$ the *trace norm* of a matrix, that is, the sum of the singular values of the matrix. The trace norm is also referred to as the *nuclear norm* or the *Schatten 1-norm* in the literature. Furthermore, let us denote by $||\cdot||_\mathcal{F}$ the Frobenius norm of a matrix, defined as

$$||A||_\mathcal{F} = \sqrt{\mathrm{Tr} A A^*} = \sqrt{\sum_{i,j} |A_{ij}|^2} \,. \tag{1}$$

The Frobenius norm is identical to the *Schatten 2-norm* of a matrix, i.e. the $\ell^2$ norm of the singular value vector of the matrix. The following lemma provides a variational characterization of the trace norm in terms of the Frobenius norm.

**Lemma 1** (Jameson (1987); Ciliberto et al. (2017)). *Let $W$ be an $m \times n$ matrix and denote by $\sigma$ its vector of singular values. Then*

$$||W||_{\mathcal{T}} := \sum_{i=1}^{\min(m,n)} \sigma_i(W) = \min \frac{1}{2} \left( ||U||_{\mathcal{F}}^2 + ||V||_{\mathcal{F}}^2 \right) , \qquad (2)$$

*where the minimum is taken over all $U : m \times \min(m, n)$ and $V : \min(m, n) \times n$ such that $W = UV$. Furthermore, if $W = \tilde{U} \Sigma \tilde{V}^*$ is a singular value decomposition of $W$, then equality holds in* (2) *for the choice $U = \tilde{U}\sqrt{\Sigma}$ and $V = \sqrt{\Sigma}\tilde{V}^*$.*

The procedure to take advantage of this characterization is as follows. First, for each large GEMM in the model, replace the $m \times n$ weight matrix $W$ by the product $W = UV$ where $U : m \times \min(m, n)$ and $V : \min(m, n) \times n$. Second, replace the original loss function $\ell(W)$ by

$$\ell(UV) + \frac{1}{2}\lambda \left( ||U||_{\mathcal{F}}^2 + ||V||_{\mathcal{F}}^2 \right) . \qquad (3)$$

where $\lambda$ is a hyperparameter controlling the strength of the approximate trace norm regularization. Proposition 1 in Ciliberto et al. (2017) guarantees that minimizing the modified loss equation (3) is equivalent to minimizing the actual trace norm regularized loss:

$$\ell(W) + \lambda ||W||_{\mathcal{T}} . \qquad (4)$$

In Section 3.2.1 we show empirically that use of the modified loss (3) is indeed highly effective at reducing the trace norm of the weight matrices.

To summarize, we propose the following basic training scheme:

- **Stage 1:**
    - For each large GEMM in the model, replace the $m \times n$ weight matrix $W$ by the product $W = UV$ where $U : m \times r$, $V : r \times n$, and $r = \min(m, n)$.
    - Replace the original loss function $\ell(W)$ by

    $$\ell(UV) + \frac{1}{2}\lambda \left( ||U||_{\mathcal{F}}^2 + ||V||_{\mathcal{F}}^2 \right) , \qquad (5)$$

    where $\lambda$ is a hyperparameter controlling the strength of the trace norm regularization.
    - Train the model to convergence.
- **Stage 2:**
    - For the trained model from stage 1, recover $W = UV$ by multiplying the two trained matrices $U$ and $V$.
    - Train low rank models warmstarted from the truncated SVD of $W$. By varying the number of singular values retained, we can control the parameter versus accuracy trade-off.

One modification to this is described in Section 3.2.3, where we show that it is actually not necessary to train the stage 1 model to convergence before switching to stage 2. By making the transition earlier, training time can be substantially reduced.

## 3.2 EXPERIMENTS AND RESULTS

We report here the results of our experiments related to trace norm regularization. Our baseline model is a forward-only Deep Speech 2 model, and we train and evaluate on the widely used Wall Street Journal (WSJ) speech corpus. Except for a few minor modifications described in Appendix B, we follow closely the original paper describing this architecture (Amodei et al., 2016), and we refer the reader to that paper for details on the inputs, outputs, exact layers used, training methodology, and so on. For the purposes of this paper, suffice it to say that the parameters and computation are dominated by three GRU layers and a fully connected layer. It is these four layers that we compress through low-rank factorization. As described in Appendix B.2, in our factorization scheme, each

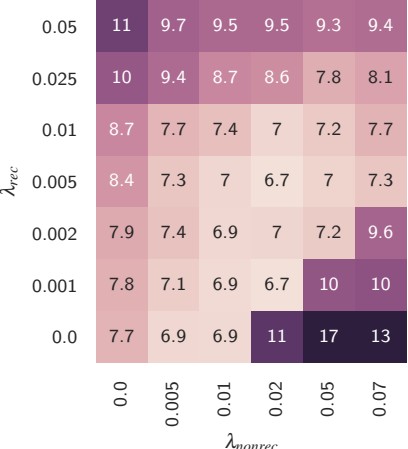 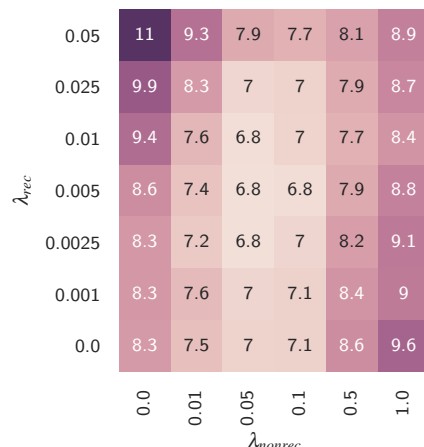

Figure 1: CER dependence on $\lambda_{rec}$ and $\lambda_{nonrec}$ for trace norm regularization (left) and $\ell^2$ regularization (right).

GRU layer involves two matrix multiplications: a *recurrent* and a *non-recurrent* one. For a simple recurrent layer, we would write

$$h_t = f(W_{nonrec}x_t + W_{rec}h_{t-1}). \tag{6}$$

For a GRU layer, there are also weights for reset and update gates, which we group with the *recurrent* matrix. See Appendix B.2 for details and the motivation for this split.

Since our work focuses only on compressing acoustic models and not language models, the error metric we report is the character error rate (CER) rather than word error rate (WER). As the size and latency constraints vary widely across devices, whenever possible we compare techniques by comparing their accuracy versus number of parameter trade-off curves. All CERs reported here are computed on a validation set separate from the training set.

### 3.2.1 STAGE 1 EXPERIMENTS

In this section, we investigate the effects of training with the modified loss function in (3). For simplicity, we refer to this as *trace norm regularization*.

As the WSJ corpus is relatively small at around 80 hours of speech, models tend to benefit substantially from regularization. To make comparisons more fair, we also trained unfactored models with an $\ell^2$ regularization term and searched the hyperparameter space just as exhaustively.

For both trace norm and $\ell^2$ regularization, we found it beneficial to introduce separate $\lambda_{rec}$ and $\lambda_{nonrec}$ parameters for determining the strength of regularization for the recurrent and non-recurrent weight matrices, respectively. In addition to $\lambda_{rec}$ and $\lambda_{nonrec}$ in initial experiments, we also roughly tuned the learning rate. Since the same learning rate was found to be optimal for nearly all experiments, we just used that for all the experiments reported in this section. The dependence of final CER on $\lambda_{rec}$ and $\lambda_{nonrec}$ is shown in Figure 1. Separate $\lambda_{rec}$ and $\lambda_{nonrec}$ values are seen to help for both trace norm and $\ell^2$ regularization. However, for trace norm regularization, it appears better to fix $\lambda_{rec}$ as a multiple of $\lambda_{nonrec}$ rather than tuning the two parameters independently.

The first question we are interested in is whether our modified loss (3) is really effective at reducing the trace norm. As we are interested in the relative concentration of singular values rather than their absolute magnitudes, we introduce the following nondimensional metric.

**Definition 1.** *Let $W$ be a nonzero $m \times n$ matrix with $d = \min(m,n) \geq 2$. Denote by $\sigma$ the $d$-dimensional vector of singular values of $W$. Then we define the nondimensional trace norm coefficient of $W$ as follows:*

$$\nu(W) := \frac{\frac{||\sigma||_{\ell^1}}{||\sigma||_{\ell^2}} - 1}{\sqrt{d} - 1}. \tag{7}$$

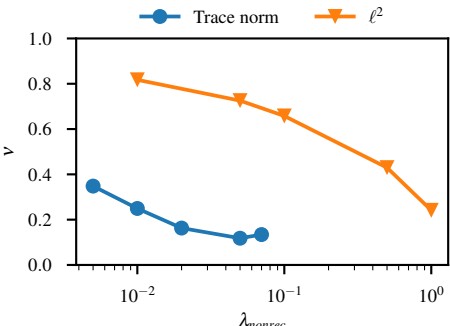 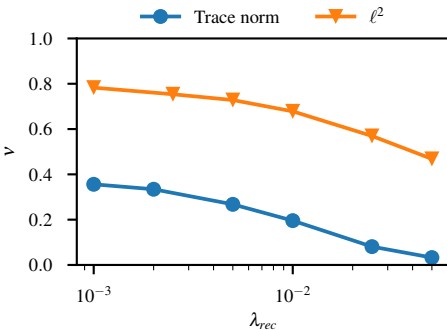

Figure 2: Nondimensional trace norm coefficient versus strength of regularization by type of regularization used during training. On the left are the results for the non-recurrent weight of the third GRU layer, with $\lambda_{rec} = 0$. On the right are the results for the recurrent weight of the third GRU layer, with $\lambda_{nonrec} = 0$. The plots for the other weights are similar.

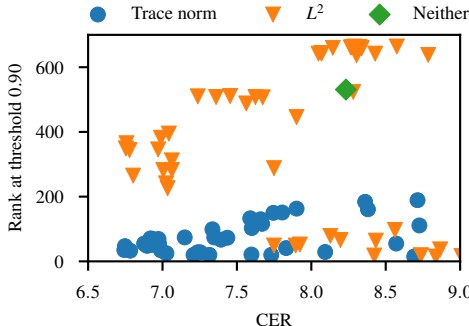 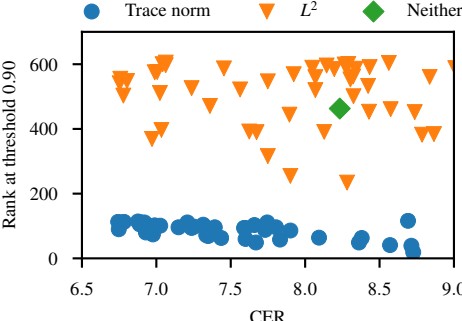

Figure 3: The truncated SVD rank required to explain 90 % of the variance of the weight matrix versus CER by type of regularization used during training. Shown here are results for the non-recurrent (left) and recurrent (right) weights of the third GRU layer. The plots for the other weights are similar.

We show in Appendix A that $\nu$ is scale-invariant and ranges from 0 for rank 1 matrices to 1 for maximal-rank matrices with all singular values equal. Intuitively, the smaller $\nu(W)$, the better $W$ can be approximated by a low rank matrix.

As shown in Figure 2, trace norm regularization is indeed highly effective at reducing the nondimensional trace norm coefficient compared to $\ell^2$ regularization. At very high regularization strengths, $\ell^2$ regularization also leads to small $\nu$ values. However, from Figure 1 it is apparent that this comes at the expense of relatively high CERs. As shown in Figure 3, this translates into requiring a much lower rank for the truncated SVD to explain, say, 90 % of the variance of the weight matrix for a given CER. Although a few $\ell^2$-regularized models occasionally achieve low rank, we observe this only at relatively high CER's and only for some of the weights. Note also that some form of regularization is very important on this dataset. The unregularized baseline model (the green points in Figure 3) achieves relatively low CER.

### 3.2.2 STAGE 2 EXPERIMENTS

In this section, we report the results of stage 2 experiments warmstarted from either trace norm or $L^2$ regularized stage 1 models.

For each regularization type, we took the three best stage 1 models (in terms of final CER: all were below 6.8) and used the truncated SVD of their weights to initialize the weights of stage 2 models. By varying the threshold of variance explained for the SVD truncation, each stage 1 model

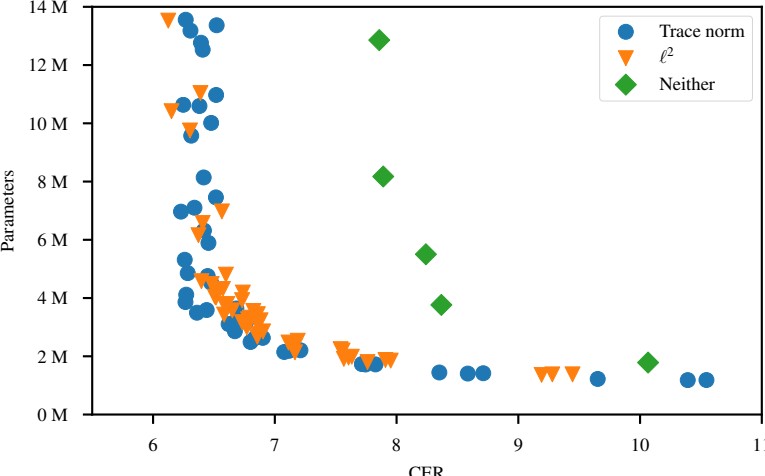

Figure 4: Number of parameters versus CER of stage 2 models colored by the type of regularization used for training the stage 1 model.

resulted into multiple stage 2 models. The stage 2 models were trained without regularization (i.e., $\lambda_{rec} = \lambda_{nonrec} = 0$) and with the initial learning rate set to three times the final learning rate of the stage 1 model.

As shown in Figure 4, the best models from either trace norm or $L^2$ regularization exhibit similar accuracy versus number of parameter trade-offs. For comparison, we also warmstarted some stage 2 models from an unregularized stage 1 model. These models are seen to have significantly lower accuracies, accentuating the need for regularization on the WSJ corpus.

### 3.2.3 REDUCING TRAINING TIME

In the previous sections, we trained the stage 1 models for 40 epochs to full convergence and then trained the stage 2 models for another 40 epochs, again to full convergence. Since the stage 2 models are drastically smaller than the stage 1 models, it takes less time to train them. Hence, shifting the stage 1 to stage 2 transition point to an earlier epoch could substantially reduce training time. In this section, we show that it is indeed possible to do so without hurting final accuracy.

Specifically, we took the stage 1 trace norm and $\ell^2$ models from Section 3.2.1 that resulted in the best stage 2 models in Section 3.2.2. In that section, we were interested in the parameters vs accuracy trade-off and used each stage 1 model to warmstart a number of stage 2 models of different sizes. In this section, we instead set a fixed target of 3 M parameters and a fixed overall training budget of 80 epochs but vary the stage 1 to stage 2 transition epoch. For each of the stage 2 runs, we initialize the learning rate with the learning rate of the stage 1 model at the transition epoch. So the learning rate follows the same schedule as if we had trained a single model for 80 epochs. As before, we disable all regularization for stage 2.

The $\ell^2$ stage 1 model has 21.7 M parameters, whereas the trace norm stage 1 model at 29.8 M parameters is slightly larger due to the factorization. Since the stage 2 models have roughly 3 M parameters and the training time is approximately proportional to the number of parameters, stage 2 models train about 7x and 10x faster, respectively, than the $\ell^2$ and trace norm stage 1 models. Consequently, large overall training time reductions can be achieved by reducing the number of epochs spent in stage 1 for both $\ell^2$ and trace norm.

The results are shown in Figure 5. Based on the left panel, it is evident that we can lower the transition epoch number without hurting the final CER. In some cases, we even see marginal CER improvements. For transition epochs of at least 15, we also see slightly better results for trace norm than $\ell^2$. In the right panel, we plot the convergence of CER when the transition epoch is 15. We find that the trace norm model's CER is barely impacted by the transition whereas the $\ell^2$ models see a

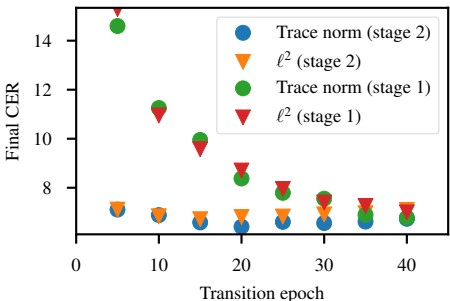 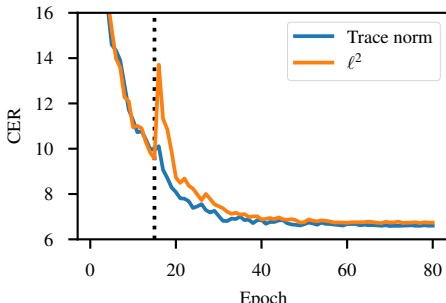

Figure 5: *Left:* CER versus transition epoch, colored by the type of regularization used for training the stage 1 model. *Right:* CER as training progresses colored by the type of regularization used in stage 1. The dotted line indicates the transition epoch.

Table 1: WER of three tiers of low rank speech recognition models and a production server model on an internal test set. This table illustrates the effect of shrinking just the acoustic model. The same large server-grade language model was used for all rows.

| Model | Parameters (M) | WER | % Relative |
|---|---|---|---|
| baseline | 115.5 | 8.78 | 0.0% |
| tier-1 | 14.9 | 9.25 | $-5.4\%$ |
| tier-2 | 10.9 | 9.80 | $-11.6\%$ |
| tier-3* | 14.7 | 9.92 | $-13.0\%$ |

*\* The tier-3 model is larger but faster than the tier-2 model. See main text for details.*

huge jump in CER at the transition epoch. Furthermore, the plot suggests that a total of 60 epochs may have sufficed. However, the savings from reducing stage 2 epochs are negligible compared to the savings from reducing the transition epoch.

## 4 APPLICATION TO PRODUCTION-GRADE EMBEDDED SPEECH RECOGNITION

With low rank factorization techniques similar[2] to those described in Section 3, we were able to train large vocabulary continuous speech recognition (LVCSR) models with acceptable numbers of parameters and acceptable loss of accuracy compared to a production server model (baseline). Table 1 shows the baseline along with three different compressed models with much lower number of parameters. The tier-3 model employs the techniques of Sections B.4 and B.3. Consequently, it runs significantly faster than the tier-1 model, even though they have a similar number of parameters. Unfortunately, this comes at the expense of some loss in accuracy.

Although low rank factorization significantly reduces the overall computational complexity of our LVCSR system, we still require further optimization to achieve real-time inference on mobile or embedded devices. One approach to speeding up the network is to use low-precision 8-bit integer representations for weight matrices and matrix multiplications (the GEMM operation in BLAS terminology). This type of quantization after training reduces both memory as well as computation requirements of the network while only introducing 2% to 4% relative increase in WER. Quantization for embedded speech recognition has also been previously studied in (Alvarez et al., 2016; Vanhoucke et al., 2011), and it may be possible to reduce the relative WER increase by quantizing the forward passes during training (Alvarez et al., 2016). As the relative WER losses from compressing the acoustic and language models were much larger for us, we did not pursue this further for the present study.

---

[2]This work was done prior to the development of our trace norm regularization. Due to long training cycles for the 10,000+ hours of speech used in this section, we started from pretrained models. However, the techniques in this section are entirely agnostic to such differences.

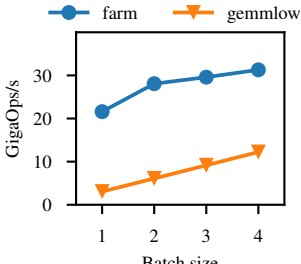 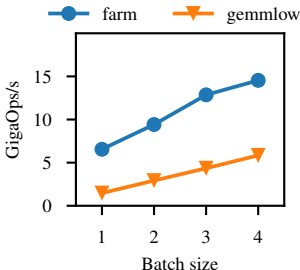 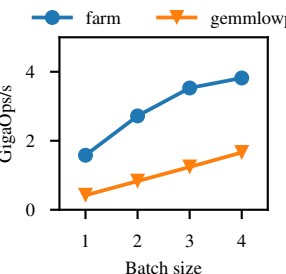

Figure 6: Comparison of our kernels (*farm*) and the gemmlowp library for matrix multiplication on iPhone 7 (left), iPhone 6 (middle), and Raspberry Pi 3 (right). The benchmark computes $Ax = b$ where $A$ is a random matrix with dimension $6144 \times 320$, and $x$ is a random matrix with dimension $320 \times$ batch size. All matrices are in unsigned 8-bit integer format.

To perform low precision matrix multiplications, we originally used the *gemmlowp* library, which provides state-of-the-art low precision GEMMs using unsigned 8-bit integer values (Jacob & Warden, 2015–2017). However, gemmlowp's approach is not efficient for small batch sizes. Our application, LVCSR on embedded devices with single user, is dominated by low batch size GEMMs due to the sequential nature of recurrent layers and latency constraints. This can be demonstrated by looking at a simple RNN cell which has the form:

$$h_t = f(Wx_t + Uh_{t-1}) \tag{8}$$

This cell contains two main GEMMs: The first, $Uh_{t-1}$, is sequential and requires a GEMM with batch size 1. The second, $Wx_t$, can in principle be performed at higher batch sizes by batching across time. However, choosing a too large batch sizes can significantly delay the output, as the system needs to wait for more future context. In practice, we found that batch sizes higher than around 4 resulted in too high latencies, negatively impacting user experience.

This motivated us to implement custom assembly kernels for the 64-bit ARM architecture (AArch64, also known as ARMv8 or ARM64) to further improve the performance of the GEMMs operations. We do not go through the methodological details in this paper. Instead, we are making the kernels and implementation details available at `https://github.com/paddlepaddle/farm`.

Figure 6 compares the performance of our implementation (denoted by *farm*) with the gemmlowp library for matrix multiplication on iPhone 7, iPhone 6, and Raspberry Pi 3 Model B. The farm kernels are significantly faster than their gemmlowp counterparts for batch sizes 1 to 4. The peak single-core theoretical performance for iPhone 7, iPhone 6, and Raspberry Pi 3 are 56.16, 22.4 and 9.6 Giga Operations per Second, respectively. The gap between the theoretical and achieved values are mostly due to kernels being limited by memory bandwidth. For a more detailed analysis, we refer to the farm website.

In addition to low precision representation and customized ARM kernels, we explored other approaches to speed up our LVCSR system. These techniques are described in Appendix B.

Finally, by combining low rank factorization, some techniques from Appendix B, int8 quantization and the farm kernels, as well as using smaller language models, we could create a range of speech recognition models suitably tailored to various devices. These are shown in Table 2.

## 5 CONCLUSION

We worked on compressing and reducing the inference latency of LVCSR speech recognition models. To better compress models, we introduced a trace norm regularization technique and demonstrated its potential for faster training of low rank models on the WSJ speech corpus. To reduce latency at inference time, we demonstrated the importance of optimizing for low batch sizes and released optimized kernels for the ARM64 platform. Finally, by combining the various techniques

Table 2: Embedded speech recognition models.

| Device | Acoustic model | Language model size (MB) | WER | % Relative | Speedup over real-time | % time spent in acoustic model |
|---|---|---|---|---|---|---|
| GPU server | baseline | 13 764 | 8.78 | 0.0% | 10.39x | 70.8% |
| iPhone 7 | tier-1 | 56 | 10.50 | −19.6% | 2.21x | 65.2% |
| iPhone 6 | tier-2 | 32 | 11.19 | −27.4% | 1.13x | 75.5% |
| Raspberry Pi 3 | tier-3 | 14 | 12.08 | −37.6% | 1.08x | 86.3% |

in this paper, we demonstrated an effective path towards production-grade on-device speech recognition on a range of embedded devices.

ACKNOWLEDGMENTS

We would like to thank Gregory Diamos, Christopher Fougner, Atul Kumar, Julia Li, Sharan Narang, Thuan Nguyen, Sanjeev Satheesh, Richard Wang, Yi Wang, and Zhenyao Zhu for their helpful comments and assistance with various parts of this paper. We also thank anonymous referees for their comments that greatly improved the exposition and helped uncover a mistake in an earlier version of this paper.

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

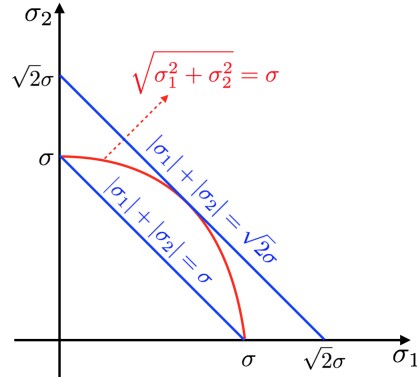

Figure 7: Contours of $||\sigma||_{\ell^1}$ and $||\sigma||_{\ell^2}$. $||\sigma||_{\ell^2}$ is kept constant at $\sigma$. For this case, $||\sigma||_{\ell^1}$ can vary from $\sigma$ to $\sqrt{2}\sigma$.

## A    NONDIMENSIONAL TRACE NORM COEFFICIENT

In this section, we describe some of the properties of the non-dimensional trace norm coefficient defined in Section 3.1.

**Proposition 1.** *Let $W, d, \sigma$ be as in Definition 1. Then*

*(i)  $\nu(cW) = \nu(W)$ for all scalars $c \in \mathbb{R} \setminus \{0\}$.*

*(ii)  $0 \le \nu(W) \le 1$.*

*(iii)  $\nu(W) = 0$ if and only if $W$ has rank 1.*

*(iv)  $\nu(W) = 1$ if and only if $W$ has maximal rank and all singular values are equal.*

*Proof.* Since we are assuming $W$ is nonzero, at least one singular value is nonzero and hence $||\sigma||_{\ell^2} \ne 0$. Property (i) is immediate from the scaling property $||c\sigma|| = |c| \cdot ||\sigma||$ satisfied by all norms.

To establish the other properties, observe that we have

$$(\sigma_i + \sigma_j)^2 \ge \sigma_i^2 + \sigma_j^2 \ge 2\left(\frac{1}{2}\sigma_i + \frac{1}{2}\sigma_j\right)^2. \tag{9}$$

The first inequality holds since singular values are nonnegative, and the inequality is strict unless $\sigma_i$ or $\sigma_j$ vanishes. The second inequality comes from an application of Jensen's inequality and is strict unless $\sigma_i = \sigma_j$. Thus, replacing $(\sigma_i, \sigma_j)$ by $(\sigma_i + \sigma_j, 0)$ preserves $||\sigma||_{\ell^1}$ while increasing $||\sigma||_{\ell^2}$ unless one of $\sigma_i$ or $\sigma_j$ is zero. Similarly, replacing $(\sigma_i, \sigma_j)$ by $(\frac{1}{2}\sigma_i + \frac{1}{2}\sigma_j, \frac{1}{2}\sigma_i + \frac{1}{2}\sigma_j)$ preserves $||\sigma||_{\ell^1}$ while decreasing $||\sigma||_{\ell^2}$ unless $\sigma_i = \sigma_j$. By a simple argument by contradiction, it follows that the minima occur for $\sigma = (\sigma_1, 0, \ldots, 0)$, in which case $\nu(W) = 0$ and the maxima occur for $\sigma = (\sigma_1, \ldots, \sigma_1)$, in which case $\nu(W) = 1$. □

We can also obtain a better intuition about the minimum and maximum of $\nu(W)$ by looking at the 2D case visualized in Figure 7. For a fixed $||\sigma||_{\ell^2} = \sigma$, $||\sigma||_{\ell^1}$ can vary from $\sigma$ to $\sqrt{2}\sigma$. The minimum $||\sigma||_{\ell^1}$ happens when either $\sigma_1$ or $\sigma_2$ are zero. For these values $||\sigma||_{\ell^2} = ||\sigma||_{\ell^1}$ and as a result $\nu(W) = 0$. Similarly, the maximum $||\sigma||_{\ell^1}$ happens for $\sigma_1 = \sigma_2$, resulting in $\nu(W) = 1$.

## B    MODEL DESIGN CONSIDERATIONS

We describe here a few preliminary insights that informed our choice of baseline model for the experiments reported in Sections 3 and 4.

Since the target domain is on-device streaming speech recognition with low latency, we chose to focus on Deep Speech 2 like models with forward-only GRU layers (Amodei et al., 2016).

## B.1 GROWING RECURRENT LAYER SIZES

Across several data sets and model architectures, we consistently found that the sizes of the recurrent layers closer to the input could be shrunk without affecting accuracy much. A related phenomenon was observed in Prabhavalkar et al. (2016): When doing low rank approximations of the acoustic model layers using SVD, the rank required to explain a fixed threshold of explained variance grows with distance from the input layer.

To reduce the number of parameters of the baseline model and speed up experiments, we thus chose to adopt growing GRU dimensions. Since the hope is that the compression techniques studied in this paper will automatically reduce layers to a near-optimal size, we chose to not tune these dimensions, but simply picked a reasonable affine increasing scheme of 768, 1024, 1280 for the GRU dimensions, and dimension 1536 for the final fully connected layer.

## B.2 PARAMETER SHARING IN THE LOW RANK FACTORIZATION

For the recurrent layers, we employ the Gated Recurrent Unit (GRU) architecture proposed in Cho et al. (2014); Chung et al. (2014), where the hidden state $h_t$ is computed as follows:

$$
\begin{aligned}
z_t &= \sigma(W_z x_t + U_z h_{t-1} + b_z) \\
r_t &= \sigma(W_r x_t + U_r h_{t-1} + b_r) \\
\tilde{h}_t &= f(W_h x_t + r_t \cdot U_h h_{t-1} + b_h) \\
h_t &= (1 - z_t) \cdot h_{t-1} + z_t \cdot \tilde{h}_t
\end{aligned}
\tag{10}
$$

where $\sigma$ is the sigmoid function, $z$ and $r$ are update and reset gates respectively, $U_z, U_r, U_h$ are the three recurrent weight matrices, and $W_z, W_r, W_h$ are the three non-recurrent weight matrices.

We consider here three ways of performing weight sharing when doing low rank factorization of the 6 weight matrices.

1. **Completely joint factorization.** Here we concatenate the 6 weight matrices along the first dimension and apply low rank factorization to this single combined matrix.
2. **Partially joint factorization.** Here we concatenate the 3 recurrent matrices into a single matrix $U$ and likewise concatenate the 3 non-recurrent matrices into a single matrix $W$. We then apply low rank factorization to each of $U$ and $W$ separately.
3. **Completely split factorization.** Here we apply low rank factorization to each of the 6 weight matrices separately.

In (Prabhavalkar et al., 2016; Kuchaiev & Ginsburg, 2017), the authors opted for the LSTM analog of *completely joint factorization*, as this choice has the most parameter sharing and thus the highest potential for compression of the model. However, we decided to go with *partially joint factorization* instead, largely for two reasons. First, in pilot experiments, we found that the $U$ and $W$ matrices behave qualitatively quite differently during training. For example, on large data sets the $W$ matrices may be trained from scratch in factored form, whereas factored $U$ matrices need to be either warmstarted via SVD from a trained unfactored model or trained with a significantly lowered learning rate. Second, the $U$ and $W$ split is advantageous in terms of computational efficiency. For the non-recurrent $W$ GEMM, there is no sequential time dependency and thus its inputs $x$ may be batched across time.

Finally, we compared the partially joint factorization to the completely split factorization and found that the former indeed led to better accuracy versus number of parameters trade-offs. Some results from this experiment are shown in Table 3.

## B.3 MEL AND SMALLER CONVOLUTION FILTERS

Switching from 161-dimensional linear spectrograms to 80-dimensional mel spectrograms reduces the per-timestep feature dimension by roughly a factor of 2. Furthermore, and likely owing to this

Table 3: Performance of completely split versus partially joint factorization of recurrent weights.

| | Completely split | | Partially joint | |
|---|---|---|---|---|
| SVD threshold | Parameters (M) | CER | Parameters (M) | CER |
| 0.50 | 6.3 | 10.3 | 5.5 | 10.3 |
| 0.60 | 8.7 | 10.5 | 7.5 | 10.2 |
| 0.70 | 12.0 | 10.3 | 10.2 | 9.9 |
| 0.80 | 16.4 | 10.1 | 13.7 | 9.7 |

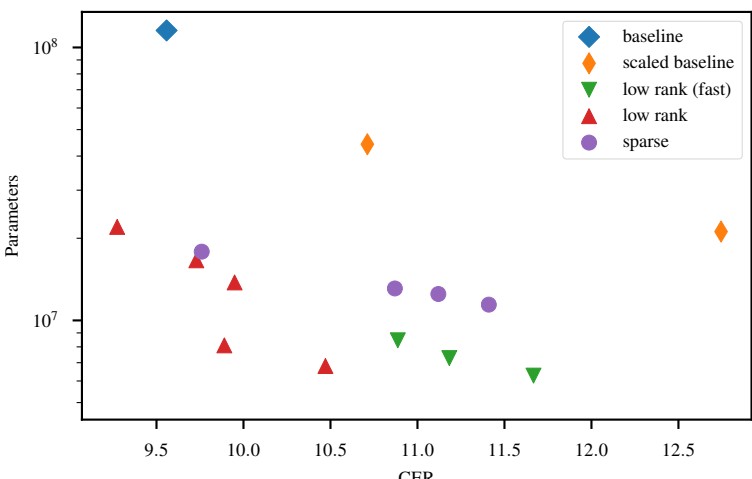

Figure 8: CER versus parameter on an internal dataset, colored by parameter reduction technique.

switch, we could reduce the frequency-dimension size of the convolution filters by a factor of 2. In combination, this means about a 4x reduction in compute for the first and second convolution layers, and a 2x reduction in compute for the first GRU layer.

On the WSJ corpus as well as an internal dataset of around 1,000 hours of speech, we saw little impact on accuracy from making this change, and hence we adopted it for all experiments in Section 3.

## B.4 GRAM-CTC AND INCREASED STRIDE IN CONVOLUTIONS

Gram-CTC is a recently proposed extension to CTC for training models that output variable-size grams as opposed to single characters (Liu et al., 2017). Using Gram-CTC, we were able to increase the time stride in the second convolution layer by a factor of 2 with little to no loss in CER, though we did have to double the number of filters in that same convolution layer to compensate. The net effect is a roughly 2x speedup for the second and third GRU layers, which are the largest. This speed up more than makes up for the size increase in the softmax layer and the slightly more complex language model decoding when using Gram-CTC. However, for a given target accuracy, we found that Gram-CTC models could not be shrunk as much as CTC models by means of low rank factorization. That is, the net effect of this technique is to increase model size in exchange for reduced latency.

## B.5 LOW RANK FACTORIZATION VERSUS LEARNED SPARSITY

Shown in Figure 8 is the parameter reduction versus relative CER increase trade-off for various techniques on an internal data set of around 1,000 hours of speech.

The baseline model is a Deep Speech 2 model with three forward-GRU layers of dimension 2560, as described in Amodei et al. (2016). This is the same baseline model used in the experiments of Narang et al. (2017), from which paper we also obtained the sparse data points in the plot. Shown also are versions of the baseline model but with the GRU dimension scaled down to 1536 and 1024. Overall, models with low rank factorizations on all non-recurrent and recurrent weight matrices are seen to provide the best CER vs parameters trade-off. All the low rank models use growing GRU dimensions and the partially split form of low rank factorization, as discussed in Sections B.1 and B.2. The models labeled *fast* in addition use Gram-CTC as described in Section B.4 and mel features and reduced convolution filter sizes as described in Section B.3.

As this was more of a preliminary comparison to some past experiments, the setup was not perfectly controlled and some models were, for example, trained for more epochs than others. We suspect that, given more effort and similar adjustments like growing GRU dimensions, the sparse models could be made competitive with the low rank models. Even so, given the computational advantage of the low rank approach over unstructured sparsity, we chose to focus only on the former going forward. This does not, of course, rule out the potential usefulness of other, more structured forms of sparsity in the embedded setting.

