# OpenReview forum: "Trace norm regularization and faster inference for embedded speech recognition RNNs"
_ICLR.cc/2018/Conference — Reject_

### Official Review · AnonReviewer2 · 2017-11-27

**Rating:** 5
**Confidence:** 5

**Review:**

The authors propose a strategy for compressing RNN acoustic models in order to deploy them for embedded applications. The technique consists of first training a model by constraining its trace norm, which allows it to be well-approximated by a truncated SVD in a second fine-tuning stage. Overall, I think this is interesting work, but I have a few concerns which I’ve listed below:

1. Section 4, which describes the experiments of compressing server sized acoustic models for embedded recognition seems a bit “disjoint” from the rest of the paper. I had a number of clarification questions spefically on this section:
- Am I correct that the results in this section do not use the trace-norm regularization at all? It would strengthen the paper significantly if the experiments presented on WSJ in the first section were also conducted on the “internal” task with more data.
- How large are the training/test sets used in these experiments (for test sets, number of words, for training sets, amount of data in hours (is this ~10,000hrs), whether any data augmentation such as multi-style training was done, etc.)
- What are the “tier-1” and “tier-2” models in this section? It would also aid readability if the various models were described more clearly in this section, with an emphasis on structure, output targets, what LMs are used, how are the LMs pruned for the embedded-size models, etc. Also, particularly given that the focus is on embedded speech recognition, of which the acoustic model is one part, I would like a few more details on how decoding was done, etc.
- The details in appendix B are interesting, and I think they should really be a part of the main paper. That being said, the results in Section B.5, as the authors mention, are somewhat preliminary, and I think the paper would be much stronger if the authors can re-run these experiments were models are trained to convergence.
- The paper focuses fairly heavily on speech recognition tasks, and I wonder if it would be more suited to a conference on speech recognition.

2. Could the authors comment on the relative training time of the models with the trace-norm regularizer, L2-regularizer and the unconstrained model in terms of convergence time.

3. Clarification question: For the WSJ experiments was the model decoded without an LM? If no LM was used, then the choice of reporting results in terms of only CER is reasonable, but I think it would be good to also report WERs on the WSJ set in either case.

4. Could the authors indicate the range of values of \lambda_{rec} and \lambda_{nonrec} that were examined in the work? Also, on a related note, in Figure 2, does each point correspond to a specific choice of these regularization parameters?

5. Figure 4: For the models in Figure 4, it would be useful to indicate the starting CER of the stage-1 model before stage-2 training to get a sense of how stage-2 training impacts performance.

6. Although the results on the WSJ set are interesting, I would be curious if the same trends and conclusions can be drawn from a larger dataset -- e.g., the internal dataset that results are reported on later in the paper, or on a set like Switchboard. I think these experiments would strengthen the paper.

7. The experiments in Section 3.2.3 were interesting, since they demonstrate that the model can be warm-started from a model that hasn’t fully converged. Could the authors also indicate the CER of the model used for initialization in addition to the final CER after stage-2 training in Figure 5.

8. In Section 4, the authors mention that quantization could be used to compress models further although this is usually degrades WER by 2--4% relative. I think the authors should consider citing previous works which have examined quantization for embedded speech recognition [1], [2]. In particular, note that [2] describes a technique for training with quantized forward passes which results in models that have smaller performance degradation relative to quantization after training.
References:
[1] Vincent Vanhoucke, Andrew Senior, and Mark Mao, “Improving the speed of neural networks on cpus,” in Deep Learning and Unsupervised Feature Learning Workshop, NIPS, 2011.
[2] Raziel Alvarez, Rohit Prabhavalkar, Anton Bakhtin, “On the efficient representation and execution of deep acoustic models,” Proc. of Interspeech, pp. 2746 -- 2750, 2016.

9. Minor comment: The authors use the term “warmstarting” to refer to the process of training NNs by initializing from a previous model. It would be good to clarify this in the text.

---

> ### Author Response · Authors · 2017-12-28
> **Explanation of content prioritization and organization**
>
> Thank you for your very thorough review and detailed feedback.
>
> Before addressing specific questions and comments, we would like to elaborate more on why we chose this venue to present our work and on the principles behind our paper’s organization. We agree that a lot of attention is being paid to the speech-specific aspects in our paper. This is natural given the title and since all of the experimental results we report are for speech recognition. However, it is our firm conviction that the technique we present as well as the techniques of Prabhavalkar et al.  are much more broadly applicable.
>
> Consequently, we had originally somewhat broader ambitions. We hoped to compare sparsity and low-rank factorization in more detail, on both speech recognition and other tasks like language modelling (on, say, Penn Treebank or the billion words corpus). Due to resource constraints, we could not run all experiments we hoped to and had to prioritize. Our organizing principle was to put what we thought was of more general interest in the main text, and shift the more preliminary work and the work that we felt was very speech-specific to appendices. We think that material is nonetheless valuable and we hope our inclusion of it can invite further research from the community to expand upon the issues raised and the solutions offered.
>
> What is in the main text of the paper, we believe could in principle be applied just as well to any other deep models involving dense or recurrent layers.
>
> Regarding your specific points:
>
> 1. As described above, it was a conscious decision to not focus too much on the speech-specific aspects in the main text of the paper. We plan to add some more details to the appendix later on, but detailed experiments we will need to relegate, as you suggested, to a possible future speech conference submission. Regarding section 4: We do wish we could have reported trace-norm-regularized results here, but in the end that would have cannibalized too many resources from other higher-priority experiments. (A single training run on our large 10,000+ hour speech datasets may occupy 16 GPU’s and take, with interruptions, around 3 to 4 weeks to complete.) As a result we introduced the newly developed kernels with only a few models we had already started training before the techniques in Section 3 were developed.
>
> 2. Unfortunately, due to training on different types of hardware with various interruptions, we could not compile meaningful wall-clock training time comparisons.
>
> 3. Correct, for WSJ we did not use a language model. As we were interested in relative performance of different compression techniques for the acoustic model only, we decided to keep the WSJ experiments as simple as possible.
>
> 4. Figure 1 shows the lambda values examined for stage 1. Yes, for Figure 2 we show the variation with respect to one of the lambda's when the other lambda is fixed at 0.
>
> 5. Thank you for the suggestion. We have fixed Figure 4 (please see our response to Reviewer 2 for further details) and the behavior for all points is more consistent now. All the stage 1 models used for warm-starting the points in this figure had below 6.8 final CER. We look at stage 1 CER vs. stage 2 CER in response to your point 7 below, where the effect is more interesting.
>
> 6. This is a great suggestion for follow-up work. Unfortunately, due to resource constraints, we could not pursue this for the present paper.
>
> 7. Great suggestion. We have updated Figure 5 to include this information. As is clearer now, the stage 1 models trained for only a few epochs are really very far from being fully converged and yet are still good enough to be used for warm-starting successful stage 2 models.
>
> 8. Good point. As the relative WER losses we saw from compressing the language and acoustic models were much larger the relative loss from quantization, we chose to not pursue quantization further for this particular study. However, as you suggest, we should at least point to the relevant literature. We have added these citations and clarified this in the text.
>
> 9. Thank you for pointing this out. We have clarified this in the text.

---

### Official Review · AnonReviewer3 · 2017-11-27
**Model compression with trace norm regularization - pertinent details on experiments missing**

**Rating:** 4
**Confidence:** 3

**Review:**

The problem considered in the paper is of compressing large networks (GRUs) for faster inference at test time.

The proposed algorithm uses a two step approach: 1)  use trace norm regularization (expressed in variational form) on dense parameter matrices at training time without constraining the number of parameters, b) initializing from the SVD of parameters trained in stage 1, learn a new network with reduced number of parameters.

The experiments on WSJ dataset are promising towards achieving a trade-off between number of parameters and accuracy.

I have the following questions regarding the experiments:
1. Could the authors confirm that the reported CERS are on validation/test dataset and not on train/dev data? It is not explicitly stated. I hope it is indeed the former, else I have a major concern with the efficacy of the algorithm as ultimately, we care about the test performance of the compressed models in comparison to uncompressed model.

2. In B.1 the authors use an increasing number units in the hidden layers of the GRUs as opposed to a fixed size like in Deep Speech 2, an obvious baseline that is missing from the  experiments is the comparison with *exact* same GRU (with  768, 1024, 1280, 1536 hidden units) *without any compression*.

3.  What do different points in Fig 3 and 4 represent. What are the values of lamdas that were used to train (the l2 and trace norm regularization) the Stage 1 of models shown in Fig 4. I want to understand what is the difference in the  two types of  behavior of orange points (some of them seem to have good compression while other do not - it the difference arising from initialization or different choice of lambdas in stage 1.

It is interesting that although L2 regularization does not lead to low \nu parameters in Stage 1, the compression stage does have comparable performance to that of trace norm minimization. The authors point it out, but a further investigation might be interesting.

Writing:
1. The GRU model for which the algorithm is proposed is not introduced until the appendix. While it is a standard network, I think the details should still be included in the main text to understand some of the notation referenced in the text like “\lambda_rec” and “\lambda_norec”

---

> ### Author Response · Authors · 2017-12-28
> **Fixed mistake in Figure 4**
>
> Thank you for your thorough review. Thanks to your comment 3 in particular, we found an error in our preparation of Figure 4. We have remedied this situation and the figure looks more reasonable now. Unfortunately, our claim of more “consistent” good results appears weakened through this finding. However, the rest of the paper is not affected by this mistake.
>
> To respond to your points in detail:
>
> 1.Yes, we did the hyperparameter comparisons on a validation set that is separate from the train set. We have clarified this in the text.
>
> 2. Figure 3 shows fully converged models that are uncompressed (as we only do the compression for stage 2). The green points correspond to the baseline model mentioned in B.1, trained without any regularization. Without regularization, this baseline model is seen to perform quite poorly in terms of final CER. For WSJ, we found that models benefit greatly from regularization. Therefore, to have fair baselines to compare trace norm regularization against, we tuned L2 regularized models just as extensively as we tuned trace norm regularized models. The L2 regularized models are the orange points. We have clarified this in the text.
>
> 3. Thank you for drawing our attention again to the orange points in Figure 4. It turns out we made an error: the stage 1 models used for Figure 4 (for both trace norm and L2 regularization) were actually selected to another criterion regarding CER vs. rank at 90% trade-off we considered earlier on, rather than just best CER as we indicated in the text. After fixing the criterion to “best CER” as we had intended, there is no longer such drastically different behavior between the orange points. We have corrected the figure and updated the claims about more consistent training.
> The corrected lambda values and the CER values of the models that were used as starting points for the stage 2 experiments are as follows:
> 	L2 models, CER, 𝜆nonrec,𝜆rec
> 	1, 6.6963, 0.05, 0.01
> 	2, 6.7536, 0.05, 0.005
> 	3, 6.7577, 0.05, 0.0025
> 	Trnorm models, CER, 𝜆nonrec,𝜆rec
> 	1, 6.6471, 0.02, 0.001
> 	2, 6.7475, 0.02, 0.005
> 	3, 6.7823, 0.02, 0.0005
>
>
> Writing: We have clarified the meaning of “rec” and “nonrec” in the main body of the text. We did not want to go into the full details of the Deep Speech 2 architecture in the main text, as we feel the details are not very pertinent to our present study and may distract the reader from the generality of the ideas. However, we have tried to provide more detail on those parts that are relevant. We hope the balance we struck now improves the exposition.

---

> > ### Comment · AnonReviewer3 · 2018-01-13
> > **Response to authors**
> >
> > After the correction in Figure 4, for final compression performance,  trace norm regularization proposed by this paper has  performance comparable to more standard L2 performance. In the light of this new experiment, there is not enough evidence to prefer using trace norm regularization and factorized weights in stage 1. In fact, the factorized representation doubles the number of parameters to be learned in stage 1.
> >
> > The experiments do not seem to validate the significance of the main contribution of paper - namely using a trace norm regularization in stage 1 for better  performance after compression with low rank factorization. Am I missing something here?

---

> > > ### Author Response · Authors · 2018-01-16
> > > **Clarification of contributions**
> > >
> > > Indeed, at this point, it seems hard to escape the conclusion that trace norm regularization is not substantially or at all superior to L2 regularization with respect to the number of parameters versus CER trade-off. In retrospect, we should have written the paper quite differently to highlight the comparison of regularization techniques as well as some of our other main contributions.
> > >
> > > For the record and for the benefit of possible future viewers of this page, we would like to list one more time some of these contributions:
> > >
> > > 1. The initial baseline we faced, based on the literature, is actually the unregularized green points in Figure 4. Producing the strong L2 regularized baseline is itself a contribution of this paper. We found that for both trace norm and L2 regularization, getting such strong results requires separate regularization strengths for the hidden-to-hidden and input-to-hidden weights of recurrent layers. See Figure 1.
> > >
> > > 2. We showed that, whether using L2 or trace norm regularization, it is not necessary to train "stage 1" models fully. A few epochs should suffice. This could substantially speed up training of large models. (See Figure 5.)
> > >
> > > 3. We created and made publicly available efficient GEMM kernels for small batch sizes on the ARM platform.
> > >
> > > Finally, as a pointer for possible future readers, we would like to mention that we do not think the approximate doubling of parameters in stage 1 using trace norm regularization is a serious obstacle. We suspect using rank r = min(m,n)/2 instead of r = min(m, n) in stage 1 would not impact results much for most problems. However, to be clear, we have not tested this and do not report this in the paper.

---

### Official Review · AnonReviewer1 · 2017-12-01
**This paper presents a trace norm regularization technique for factorized matrix multiplication with the purpose of overcoming the computational complexity in DNN and RNN**

**Rating:** 5
**Confidence:** 3

**Review:**

Paper is well written and clearly explained. The paper is a experimental paper as it has more content on the experimentation and less content on problem definition and formulation. The experimental section is strong and it has evaluated across different datasets and various scenarios. However, I feel the contribution of the paper toward the topic is incremental and not significant enough to be accepted in this venue. It only considers a slight modification into the loss function by adding a trace norm regularization.

---

> ### Author Response · Authors · 2017-12-28
> **We contend the paper is of more general interest.**
>
> We would like to thank you for taking the time to review our paper. Although these particular results for speech may appear incremental, we believe the methodologies and insights go far beyond speech recognition and should be of interest to researchers working on low rank methods and compressing large non-convolutional neural networks. In addition to the systematic modification to the loss function that we propose, we also present a methodology for training models with this modified loss function and also a methodology for studying the effectiveness and making fair comparisons of such techniques. There are also other critical insights that were necessary to make this work, such as the need to treat the recurrent (hidden-to-hidden) and non-recurrent (input-to-hidden) weights of recurrent layers separately, and regularize them with different strengths.
>
> Our goal in this paper is to lay the groundwork to attract more attention from the community and invite further study of the technique we present as well as the techniques of Prabhavalkar et al. and others that we build upon. The combination of these techniques, as we have shown, could also be potentially useful for speeding up the training of large networks. We hope that our work will promote the use of factorized matrices in the research community, resulting in a more compact representation of neural networks.

---

### Decision · Program_Chairs · 2018-01-29
**ICLR 2018 Conference Acceptance Decision**

**Decision:**

Reject

**Comment:**

Pros
-- Shows alternative strategies to train low-rank factored weight matrices for recurrent nets.

Cons
-- Minor modifications (and gains) over other forms of regularization like L2.
-- Results are only on an ASR task, so it’s not entirely clear how they’ll work on other tasks.

As pointed out by the reviewers, unless the authors show that the techniques generalize well to other tasks, and larger datasets it hard to accept it to the main conference. The AC, therefore, recommends that the paper be rejected.